# DEAD-box polypeptide 43 facilitates piRNA amplification by actively liberating RNA from Ago3-piRISC

Ryo Murakami[1] ⓘD, Tetsutaro Sumiyoshi[1], Lumi Negishi[2] & Mikiko C Siomi[1,*] ⓘD

## Abstract

The piRNA amplification pathway in *Bombyx* is operated by Ago3 and Siwi in their piRISC form. The DEAD-box protein, Vasa, facilitates Ago3-piRISC production by liberating cleaved RNAs from Siwi-piRISC in an ATP hydrolysis-dependent manner. However, the Vasa-like factor facilitating Siwi-piRISC production along this pathway remains unknown. Here, we identify DEAD-box polypeptide 43 (DDX43) as the Vasa-like protein functioning in Siwi-piRISC production. DDX43 belongs to the helicase superfamily II along with Vasa, and it contains a similar helicase core. DDX43 also contains a K-homology (KH) domain, a prevalent RNA-binding domain, within its N-terminal region. Biochemical analyses show that the helicase core is responsible for Ago3-piRISC interaction and ATP hydrolysis, while the KH domain enhances the ATPase activity of the helicase core. This enhancement is independent of the RNA-binding activity of the KH domain. For maximal DDX43 RNA-binding activity, both the KH domain and helicase core are required. This study not only provides new insight into the piRNA amplification mechanism but also reveals unique collaborations between the two domains supporting DDX43 function within the pathway.

**Keywords** *Bombyx*; DDX43; DEAD-box; KH domain; piRNA
**Subject Category** RNA Biology

## Introduction

PIWI-interacting RNAs (piRNAs) are small non-coding RNAs enriched within the germline where they repress transposons to maintain genome integrity (Czech *et al*, 2018; Yamashiro & Siomi, 2018). To achieve this, piRNAs assemble into piRNA-induced silencing complexes (piRISC) along with PIWI proteins and target these complexes toward transposon transcripts through RNA–RNA base-pairing (Reuter *et al*, 2011; Zhang *et al*, 2018). The loss-of-function of piRISC leads to transposon derepression and mobile transposition

within the germline genome, resulting in severe defects in gonadal development, as well as infertility (Klattenhoff & Theurkauf, 2008; Chuma & Nakano, 2013; Rojas-Ríos & Simonelig, 2018).

piRNAs originate from intergenic loci known as piRNA clusters where transposon fragments naturally accumulate in a nested fashion (Senti & Brennecke, 2010; Yamanaka *et al*, 2014). The RNA transcripts arising from these clusters are recognized as piRNA precursors and processed into mature piRNAs by specific machinery termed the primary pathway. The piRNAs are then amplified via the ping-pong cycle depending on the endonuclease (slicer) activity of the two PIWI members (Czech & Hannon, 2016; Yamashiro & Siomi, 2018; Czech *et al*, 2018). In this pathway, not only the cluster transcripts but also the transposon mRNAs are consumed as the piRNA sources; therefore, the ping-pong cycle is considered to be a transposon repression pathway.

The ping-pong cycle was first discovered in *Drosophila* and mice (Aravin *et al*, 2007; Brennecke *et al*, 2007; Gunawardane *et al*, 2007) but is highly conserved across a wide range of animal species including humans (Aravin *et al*, 2008; Shibata *et al*, 2016; Shoji *et al*, 2017; Gainetdinov *et al*, 2018; Kim *et al*, 2019). The ping-pong cycle is active only within germ cells in the germline, which hampers investigation of its mechanisms. However, insights have gradually been gained through recent studies using *Bombyx* (silkworms) as the experimental model organism because ovary-derived cultured germ cells (e.g., BmN4 cells) that innately accommodate the ping-pong cycle are available in silkworms (Kawaoka *et al*, 2009; Xiol *et al*, 2014; Nishida *et al*, 2015).

Silkworms express two PIWI members, Siwi and Ago3. Bioinformatic analysis of Siwi- and Ago3-loaded piRNAs in BmN4 cells revealed typical ping-pong signatures in their sequences: Siwi-bound piRNAs are mostly antisense, while Ago3-bound piRNAs are mostly sense to transposon mRNAs. Furthermore, Siwi- and Ago3-bound piRNAs show strong complementarities along a 10-nucleotide (nt) stretch at their 5′-ends. The loss of Siwi attenuates Ago3-piRISC production (Nishida *et al*, 2015). However, Siwi-piRISCs are still produced upon Ago3 loss, with primary piRNAs loaded onto Siwi even in the absence of Ago3. This indicates that Siwi is located higher in the ping-pong hierarchy than Ago3.

Siwi-piRISC targets and cleaves transposon mRNAs into two pieces (Nishida *et al*, 2015). The 3′-fragment is loaded onto

1  Department of Biological Sciences, Graduate School of Science, The University of Tokyo, Tokyo, Japan
2  Laboratory of Chromatin Structure and Function, Institute for Quantitative Biosciences, The University of Tokyo, Tokyo, Japan
   *Corresponding author. Tel: +81 3 5841 4386; E-mail: siomim@bs.s.u-tokyo.ac.jp

piRNA-unbound Ago3, giving rise to the Ago3-piRISC precursor (pre-Ago3-piRISC). The 5′-end of the RNA fragment is inserted into the 5′-end binding pocket within Ago3. The 3′-end contains extra bases that are not found in mature piRNAs, and these need to be removed for piRISC maturation. The factor responsible for this processing is Zucchini (Zuc), an endonuclease located on the surface of mitochondria (Nishimasu *et al*, 2012; Nishida *et al*, 2018; Izumi *et al*, 2020). Following processing, the 3′-end of the now mature piRNA is finally inserted into the 3′-end binding pocket of Ago3. Ago3-piRISC is then released into the cytosol, where it, in turn, targets and cleaves antisense transposon transcripts into two pieces. The 3′-fragment is loaded onto unbound Siwi and is processed by Zuc, producing Siwi-piRISC. Siwi and Ago3 continue these reactions reciprocally, producing a large number of piRISCs in germ cells.

Our *in vitro* assays previously demonstrated that RNAs cleaved by Siwi-piRISC remain tightly bound to the piRISC (Nishida *et al*, 2015). In contrast, under the same conditions, RNAs cleaved by Ago2-RISC are immediately released from the complex. These findings indicated that Siwi-piRISC, but not Ago2-RISC, requires an additional factor to evict the cleaved RNAs. Siwi-cleaved RNAs serve as the substrate for Ago3-bound piRNA production, while Ago2-cleaved RNAs are destined to be degraded completely to accomplish target gene silencing. Considering these contexts, the results of these previous assays were as expected.

Vasa is a DEAD-box protein of the helicase superfamily II (SFII) and has been used as a germ cell marker because of its cell-specific expression (Gustafson & Wessel, 2010). *Drosophila* genetics revealed that Vasa is necessary for piRNA biogenesis in the ovaries (Ai & Kai, 2007; Malone *et al*, 2009). However, the molecular function of Vasa along the ping-pong pathway remained unknown. In our previous study, we incubated Siwi-piRISC with Vasa *in vitro*, and following incubation, we observed that the cleaved RNAs were actively released from Siwi-piRISC in an ATP-dependent manner, although Vasa failed to liberate cleaved RNAs from Ago3-piRISC (Nishida *et al*, 2015). This suggested that another Vasa-like factor functions in the ping-pong pathway to liberate cleaved RNAs from Ago3-piRISC. However, this factor remains unidentified.

In this study, we identified DEAD-box polypeptide 43 (DDX43) as the factor liberating cleaved RNAs from Ago3-piRISC in an ATP hydrolysis-dependent manner. Similar to Vasa, DDX43 belongs to SFII and contains a helicase core consisting of RecA1 and RecA2 (Talwar *et al*, 2017). DDX43-bound Ago3 was predominantly in the piRISC form, whereas Siwi in the piRISC was mainly piRNA-unbound. Further, recombinant DDX43 released cleaved RNAs from Ago3-piRISC but not from Siwi-piRISC, and DDX43 depletion in BmN4 cells impaired Ago3-dependent Siwi-piRISC production.

DDX43 also contains a K-homology (KH) domain, one of the most prevalent RNA-binding domains (Siomi *et al*, 1994; Valverde *et al*, 2008), within its N-terminal auxiliary region. This domain structure is conserved in human DDX43 (also termed the helicase antigen gene) (Talwar *et al*, 2017; Wu *et al*, 2019). Biochemical analysis revealed that the helicase core of DDX43 was responsible for Ago3 binding and ATP hydrolysis, while the KH domain enhanced the ATPase activity of the helicase core but this enhancement was not dependent on the RNA-binding activity. Maximal RNA-binding activity of DDX43 required both the KH domain and the helicase core. A new function of the KH domain was revealed.

## Results and Discussion

DEAD-box proteins including Vasa hydrolyze ATP to release the energy required to perform dedicated functions, such as RNA unwinding, in the presence of magnesium ions ($Mg^{2+}$) (Sengoku *et al*, 2006). We hypothesized that Vasa, in the absence of $Mg^{2+}$, would remain bound to Siwi-piRISC because of the resulting failures in displacing cleaved RNAs from the piRISC. This would be in agreement with previous findings that the E339Q mutant of Vasa, which is defective in ATP hydrolysis, remains bound to Siwi-piRISC along with piRNA-unbound Ago3 (Xiol *et al*, 2014; Nishida *et al*, 2015). To test this hypothesis, we prepared BmN4 cell lysates with and without 1 mM EDTA (EDTA+ and EDTA−, respectively) and conducted immunoprecipitation assays using anti-Siwi and anti-Vasa antibodies. The EDTA− lysate was supplemented with 5 mM $Mg^{2+}$. Western blotting showed that the Vasa–Siwi interaction was much tighter in the presence of EDTA (Fig 1A). Thus, $Mg^{2+}$ chelation by EDTA enhanced the Vasa–Siwi interaction.

This finding encouraged us to pursue similar experiments using anti-Ago3 antibodies to identify a Vasa-like factor for Ago3-piRISC. Co-immunoprecipitated proteins with Ago3 (Fig 1B) were forwarded for mass spectrometry analysis. This identified 171 and 259 proteins as Ago3-interactors in the EDTA− and EDTA+ lysates, respectively (Dataset EV1). Gene ontology analysis categorized seven of the proteins in the EDTA+ dataset as DNA/RNA helicases (Fig 1B). Vasa appeared in this list but its peptide sequence match (PSM) score was the lowest among the seven proteins. Three out of the seven proteins were only present in the EDTA+ dataset (listed in red in Fig 1B).

To verify their physical interactions with Ago3, we cloned the cDNAs of these seven proteins, expressed them individually in BmN4 cells, and performed immunoprecipitation. It is noted that we did not employ Dbp-2 isoform 2 in the assays because the sequence fully overlaps with that of Dbp-2 isoform 1. Instead, we employed Bel long isoform (Bel-L) and Bel short isoform (Bel-S) because both cDNAs were available in our laboratories. DDX43 bound Ago3 more tightly than the other proteins (Fig 1C), and this interaction became even stronger upon EDTA addition to the lysate (Fig 1D). The specificity of the Ago3–DDX43 interaction was verified by pull-down assays (Fig 1E). Spindle-E (Spn-E), another DEAD-box protein essential for piRNA biogenesis (Shoji *et al*, 2009; Nishida *et al*, 2015; Andress *et al*, 2016), also bound with Ago3 but to a much lesser extent (Fig 1C). We previously showed that the depletion of Spn-E in BmN4 cells attenuates primary Siwi-piRISC production, resulting in the loss of both Siwi-piRISC and Ago3-piRISC (Nishida *et al*, 2015), excluding the possibility that Spn-E is the Vasa counterpart. We, therefore, focused on DDX43 in all further investigations.

If DDX43 is the *bona fide* factor facilitating Siwi-piRISC production in the ping-pong cycle, Ago3 in the DDX43 complex would mostly be in the piRISC form. To test this, we isolated RNAs from the DDX43 complex and probed them with DNA oligos to detect two representatives of Ago3-bound piRNAs, PiggyBac-piRNA and Yamato-piRNA (Nishida *et al*, 2015). Neither of these piRNAs has been found in the Siwi-bound piRNA pool (Nishida *et al*, 2015), indicating the high specificity of Ago3 binding. A comparison of the piRNA levels in the DDX43 complex with those present with Ago3 in the whole lysate indicated that Ago3 in the DDX43 complex was indeed predominantly in the piRISC form (Figs 1F and EV1A). Siwi

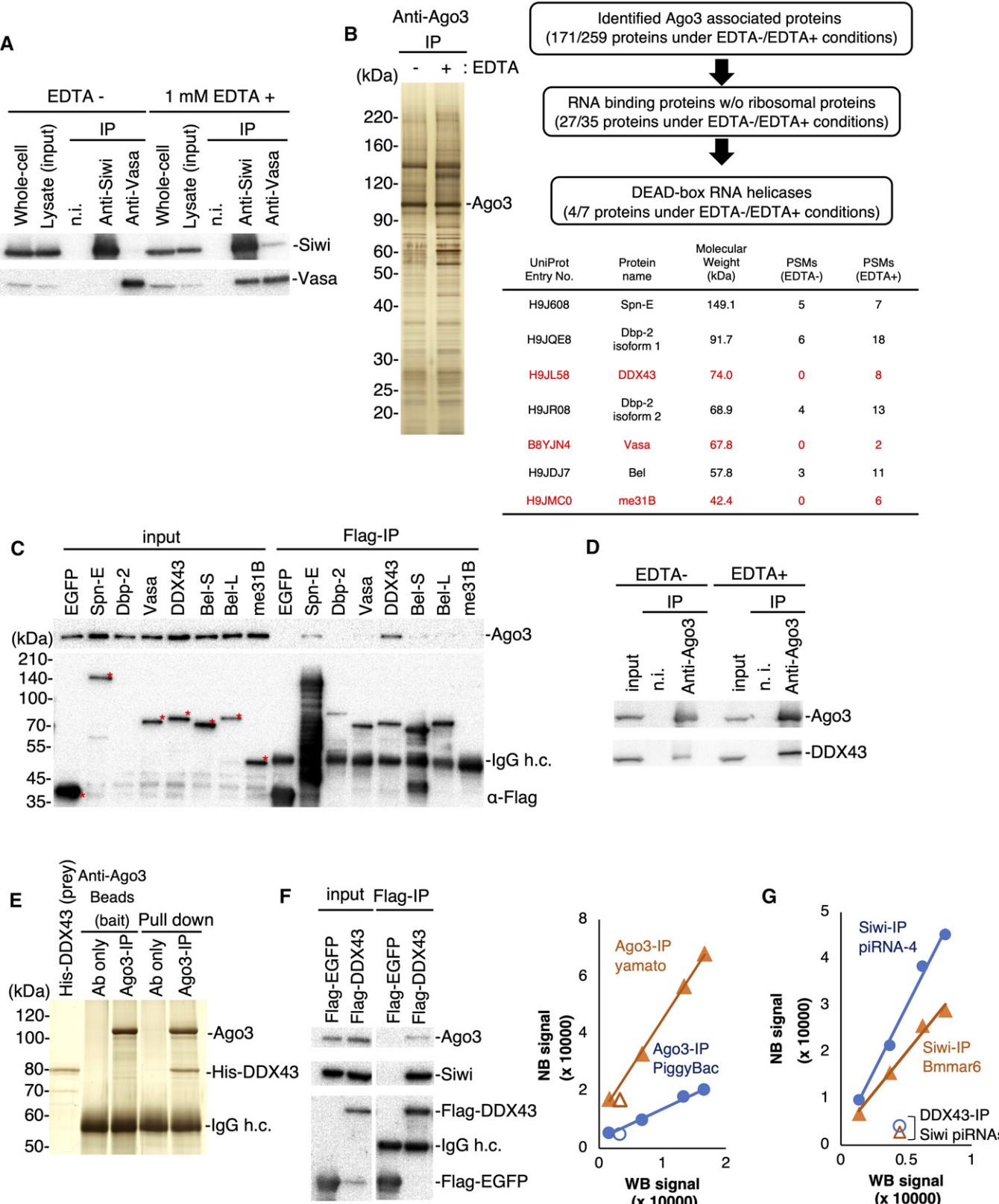

**Figure 1.**

**Figure 1.  Identification of RNA helicase DDX43 associated with Ago3.**

A    Co-immunoprecipitation of the Siwi–Vasa complex in the presence and absence of EDTA. Isolated proteins were subjected to Western blotting. n.i.: non-immune mouse IgG.

B    Co-immunoprecipitation and identification of Ago3-associated proteins. Left: The immunopurified Ago3 complex was resolved by SDS–PAGE and silver-stained. Right: DEAD-box helicases identified by mass spectrometry. Proteins identified by co-immunoprecipitation under EDTA-containing conditions only are listed in red. PSM: peptide sequence match.

C    Co-immunoprecipitation using BmN4 cells transfected with the Flag-tagged DEAD-box helicases identified by shotgun proteomics. The Flag-tagged proteins and Ago3 were detected by Western blotting. Full-length Flag-tagged proteins are indicated with red asterisks. h.c.: heavy chain.

D    Co-immunoprecipitation of endogenous Ago3 and DDX43 in the presence or absence of EDTA. These factors were detected by Western blotting.

E    Pull-down assay using high salt-washed Ago3-conjugated magnetic beads and purified His-DDX43. All samples were subjected to SDS–PAGE and detected by silver staining.

F, G   Analysis of Siwi-piRNA and Ago3-piRNA levels in Siwi and Ago3 complexes with DDX43, respectively. The WB (Western blotting) and NB (Northern blotting) signals represent the PIWI protein and piRNA levels, respectively (see also Fig EV1A and B). The relative ratios of piRISC were calculated based on the signal intensities of the PIWI proteins and piRNAs.

was also detected in the DDX43 complex (Fig 1F). Siwi appeared to be more abundant than Ago3 in the complex; however, the signal levels in this experiment vary widely depending on the titer of each antibody. Siwi in the DDX43 complex was mostly devoid of piRNA (Figs 1G and EV1B). These findings strongly support the notion that DDX43 binds Ago3-piRISC and piRNA-unbound Siwi simultaneously, just as Vasa interconnects Siwi-piRISC and piRNA-unbound Ago3 in the ping-pong cycle (Nishida *et al*, 2015).

Our recent investigations showed that the Tudor domain-containing protein, Vreteno (Vret), facilitates Ago3-dependent Siwi-piRISC production in the ping-pong cycle (Nishida *et al*, 2020). Our study also showed that a subset of nuage (germ cell-specific granules), which we termed Ago3 bodies, are central hubs in the Siwi-piRISC production. This claim was based on our observations that Ago3 bodies are not assembled in Vret-lacking cells, although Ago3-piRISC and unbound Siwi are present in these cells. In the absence of Vret, the level of Siwi-piRISC biogenesis is negligible. Further, Ago3 body assembly requires Ago3 but not Siwi.

In the present study, we examined how DDX43 depletion influences Ago3 body formation: The lack of DDX43, in contrast to Vret depletion, hardly impacted the appearance of Ago3 bodies (Fig 2A), indicating that DDX43 is dispensable for Ago3 body assembly. This suggested that DDX43 functions downstream of Vret in the pathway.

Both Vret and DDX43 interconnect Ago3-piRISC and piRNA-unbound Siwi in BmN4 cells (Nishida *et al*, 2020 and this study). We speculated that Vret and DDX43 physically interact with each other through the ping-pong cycle although such interaction would be transient. Probing the DDX43 complex components with an anti-Vret antibody showed that Vret, and particularly its longer isoform (Nishida *et al*, 2020), associated with DDX43 (Fig 2B). We then examined mutual dependencies between Vret and DDX43 in terms of Ago3 association. Vret associated with Ago3 in the absence of DDX43 (Fig 2C), agreeing with the above observation that Ago3 bodies are assembled in DDX43-depleted cells as they are in control cells (Fig 2A). Likewise, the depletion of Vret had little effect on the Ago3–DDX43 interaction (Fig 2D). This suggested the intriguing idea that Ago3 and DDX43 interact in the cytosol and co-localize to Ago3 bodies. This type of regulation would be beneficial to the ping-pong cycle because it would ensure that DDX43 immediately acts on Ago3-piRISC to liberate RNAs from the complex upon target RNA cleavage. Flag-DDX43 was detected at Ago3 bodies although the major fraction was cytosolic upon its ectopic expression in BmN4 cells (Fig 2E).

If DDX43 is the Vasa-like factor acting with Ago3-piRISC to produce Siwi-piRISC in the ping-pong cycle, piRNA loading of Ago3 should barely be impacted by the loss of DDX43, unlike the loss of Vasa (Xiol *et al*, 2014; Nishida *et al*, 2015). We knocked down Vasa and DDX43 individually in BmN4 cells and then expressed Flag-Siwi and Flag-Ago3 in these cells (Fig 3A). Examination of the PIWI-piRNA-loading status revealed that both PIWI proteins were indeed loaded fully with piRNAs in the absence of DDX43 but only Siwi was loaded with piRNAs in the absence of Vasa (Fig 3A). We then deep-sequenced Siwi-bound piRNAs in DDX43-lacking BmN4 cells and compared the resulting reads with those from Siwi-bound piRNAs in the control and Vasa-depleted cells (Nishida *et al*, 2015). In theory, Siwi-bound piRNAs in the control cells would comprise a mixture of primary and amplified piRNAs, whereas those in Vasa-depleted cells should be biased toward primary piRNAs. However, we found no obvious differences between the Siwi-bound piRNA pools before and after DDX43 depletion (Fig EV2A–C). We attributed this to residual DDX43 remaining in the cells even after extensive RNAi treatment (Fig 3A). CRISPR system-mediated DDX43 knockout would be desirable to investigate the pronounced phenotype, but was difficult to achieve in BmN4 cells for technical reasons.

We considered that another assay system would be necessary to fully elucidate the function of DDX43 within the ping-pong pathway. To this purpose, we constructed an *in vivo* assay system by modifying a system reported previously (Xiol *et al*, 2014). An artificial target RNA was designed to contain a 26-nt stretch complementary to one of the Ago3-bound piRNAs, PiggyBac-piRNA, which should, therefore, be cleaved by endogenous Ago3-piRISC upon its ectopic expression in BmN4 cells (Fig EV3A). After cleavage, the 3′-26-nt fragment was expected to bind to unloaded Siwi. This RNA loading would instantly give rise to mature Siwi-piRISC because the 26-nt fragment does not require Zuc for 3′-end processing. We first confirmed that the artificial piRNA (Art-piRNA) designed based on endogenous PiggyBac-piRNA did not exist in the naïve BmN4 cells (Fig 3B). Then, as expected, upon ectopic expression of the precursor RNA we found that Siwi was loaded with Art-piRNA (Fig 3B). The reporter plasmid lacking the complementary sequence to PiggyBac-piRNA failed to produce the Siwi–Art-piRNA complex (Fig 3B). The level of Art-piRNA loaded on Siwi was significantly reduced by the loss of Ago3 (Fig 3C), confirming the Ago3 dependency of Siwi-piRISC production. A similar result was obtained when DDX43 was depleted, although the degree of reduction was slightly less (Fig 3C). Similar results were obtained when

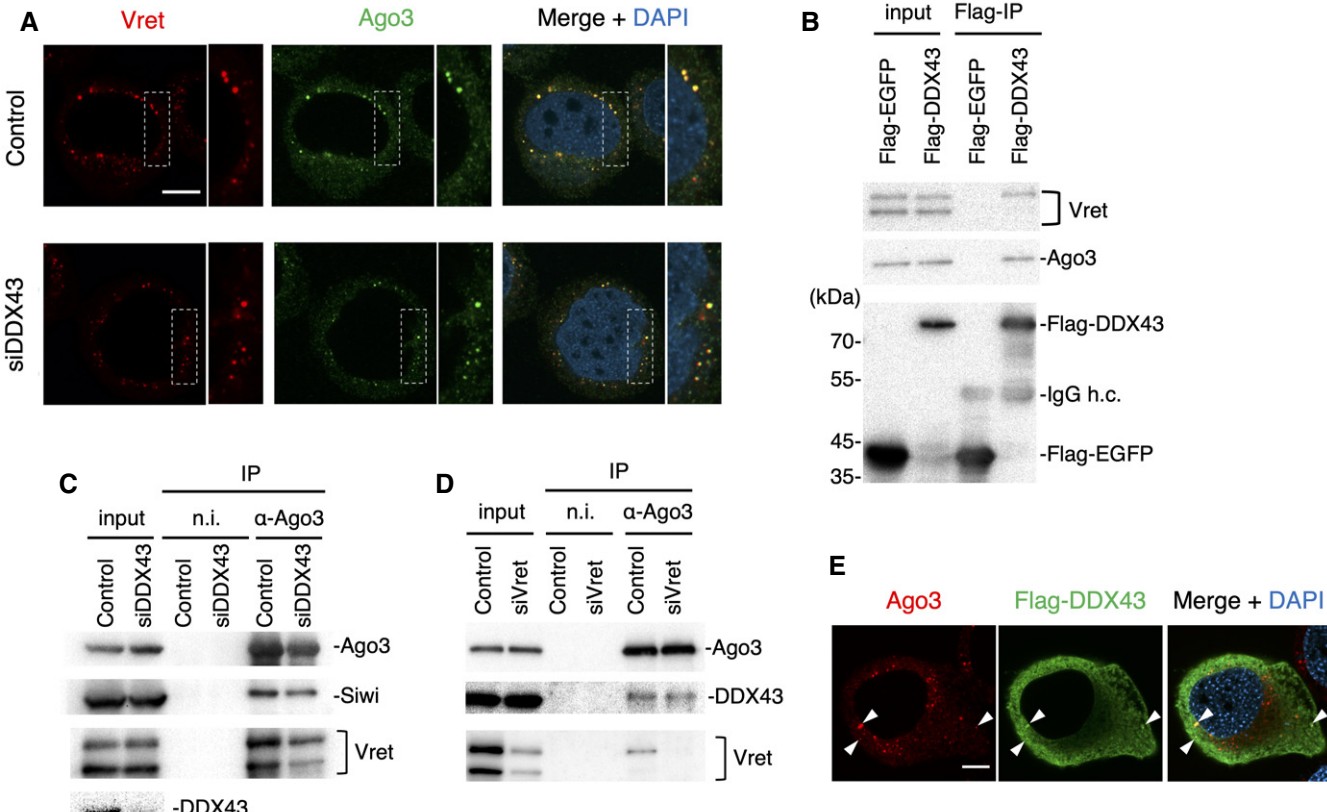

**Figure 2. DDX43 depletion does not affect the formation of the Ago3–Vret complex.**

A Immunofluorescence showing the Vret (red) and Ago3 (green) signals in DDX43-depleted BmN4 cells (siDDX43). DAPI shows the nuclei (blue). Scale bar: 10 μm. The insets show high-magnification images of the part indicated sections.

B Immunoprecipitation and Western blotting showing that DDX43 associated with Vret and Ago3.

C, D Immunoprecipitation of endogenous Siwi, DDX43, and Vret bound with Ago3 from DDX43-depleted (C) or Vret-depleted (D) cells. The factors were detected by Western blotting. Ago3 interacts with Vret and DDX43 independently of the other.

E Immunofluorescence showing Ago3 (red) and Flag-DDX43 (green) signals in BmN4 cells. DAPI shows the nuclei (blue). Arrowheads indicate colocalization of these factors. Scale bar: 10 μm.

Art-piRNAs were designed based on endogenous Yamato- and Pao-piRNAs (Fig EV3B). These results corroborate the requirement of DDX43 for Ago3-dependent Siwi-piRISC production.

We finally set out to examine whether DDX43 activity liberates RNAs from Ago3-piRISC in *in vitro* assays (Fig EV3C). Target RNAs cleaved by Ago3-piRISC remained on the complex when no other factors were added to the reaction mixture (Fig EV3D), as previously shown (Nishida *et al*, 2015). The RNA fragments were, however, released to the supernatant upon incubation with DDX43 (Fig 3D). Vasa, employed as a control, showed no such activity (Fig 3D). A time-course experiment over 1 h (Fig 3E) demonstrated that the 3′-fragment (83 nt) was liberated to the supernatant more rapidly from Ago3-piRISC (i.e., the Beads fraction) than the 5′-fragment (91 nt) in the presence of DDX43. We considered this reasonable because the 3′-fragment base-pairs with Ago3-bound piRNA over a 10-nt stretch, while the 5′-fragment base-pairs over 16 nt, which is a more stable interaction. This suggests an interesting model of "step-by-step" assembly of the Siwi-piRISC precursor (i.e., pre-Siwi-piRISC); namely, the 5′-end of 3′-fragment (i.e., the precursor for Siwi-loaded piRNA) is first bound to Siwi while the 3′-end of the precursor is still attached to Ago3-piRISC. Then, the 3′-end of

the precursor dissociates from Ago3-piRISC, fully liberating the pre-piRISC to the environment for Zuc-dependent maturation (Fig EV3E). This model was supported by the fact that after a 2-h incubation with DDX43, the longer 5′-fragment also appeared in the supernatant along with the shorter 3′-fragment (Fig 3D). Target RNAs cleaved by Siwi-piRISC remained on the complex even after incubation with DDX43 (Fig 3F), confirming the specificity of DDX43 functioning for Ago3.

Besides the central RNA helicase core, DDX43 has an N-terminal K-homology (KH) domain (Fig 4A), which is a prevalent RNA-binding domain found in many RNA-binding proteins (Valverde *et al*, 2008). This domain organization of DDX43 with an N-terminal KH domain and a central helicase core is unique and conserved at least between silkworms and humans (Fig EV4A; Talwar *et al*, 2017).

To reveal the functional contributions of the two domains in piRISC production, we produced three mutants: ΔKH, D399A, and E400Q (Fig 4A). The ΔKH mutant lacked the N-terminal 137 residues containing the complete KH domain (a.a. 77–136). The D399A and E400Q mutants contained a single amino acid substitution in the helicase core. Based on previous findings for Vasa (Xiol *et al*, 2014; Nishida *et al*, 2015), we inferred that the D399A and E400Q

mutations would abolish the ATP binding and hydrolysis activities of DDX43, respectively.

We first examined the ATP hydrolysis activity of the three mutants (Fig 4B). Strikingly, against our expectations, all mutants showed minimal ATP hydrolysis, which was in sharp contrast to the wild-type (WT) control. The ΔKH mutant contained the complete helicase core, and we, therefore, assumed that it would fully hydrolyzes ATP; but this was not the case. These findings suggested that

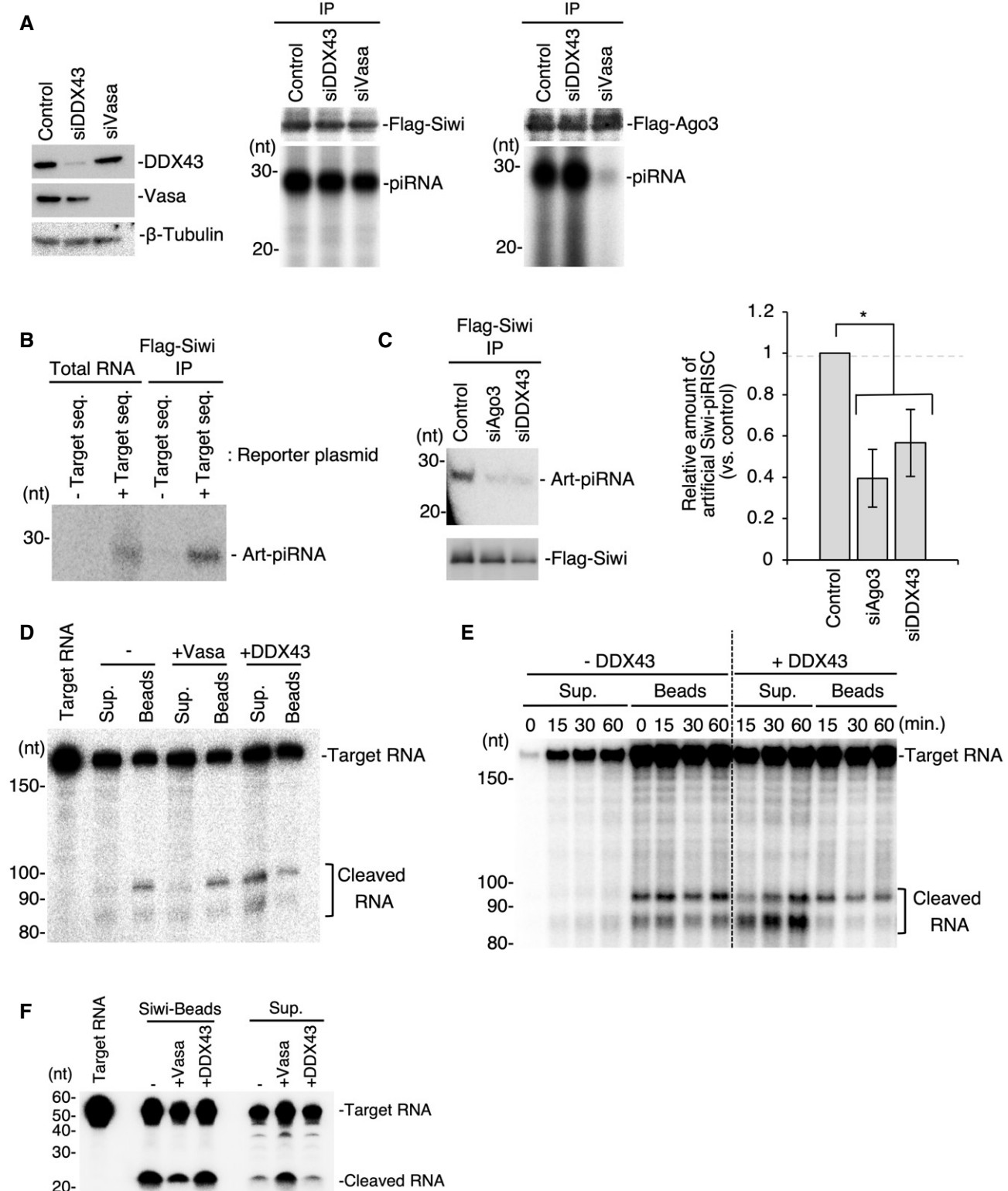

Figure 3.

**Figure 3. DDX43 unwinds Ago3-cleaved RNA and drives Siwi-piRISC biogenesis.**

A   Analysis of Siwi-piRNA and Ago3-piRNA production in DDX43- or Vasa-knockdown BmN4 cells. Left: The expression levels of DDX43 and Vasa following RNAi treatment. Middle and right: Comparison of Siwi-piRNA and Ago3-piRNA levels following DDX43 or Vasa knockdown by RNAi. The piRNAs were isolated by the immunoprecipitation of Flag-tagged proteins from knockdown cells transfected with Flag-Siwi or Flag-Ago3 and then $^{32}$P-labeled.

B   Artificial Siwi-piRNA production from a reporter plasmid encoding the PiggyBac-piRNA target sequence.

C   Analysis of the artificial piRNA levels in DDX43-knockdown BmN4 cells. Left: Artificial piRNA and Flag-Siwi were detected by Northern blotting and Western blotting, respectively. Right: Quantification of artificial Siwi-piRISC formed by the ping-pong cycle pathway. Data represent the mean ± standard deviations, $n = 3$ independent experiments. *$P < 0.05$ as determined with the *t*-test.

D   RNA-unwinding assays upon Ago3-piRISC-dependent target RNA cleavage (Fig EV3D). Recombinant Vasa or DDX43 was added and incubated in the reaction mixture upon cleavage and the materials were separated using a magnet to supernatant (Sup.) and bead (Beads) fractions. RNAs cleaved by Ago3 are shown as "Cleaved RNA". DDX43 but not Vasa released the cleaved target RNA from Ago3-piRISC.

E   Time dependency of the DDX43 unwinding activity. Sup: supernatant.

F   RNA-unwinding assays upon Ago3-piRISC-dependent target RNA cleavage as in (D). Recombinant DDX43 was added and incubated in the reaction mixture upon cleavage for 0 min (i.e., no cleavage), 15, 30, and 60 min, and the materials were separated using a magnet to supernatant (Sup.) and bead (Beads) fractions. RNAs cleaved by Ago3 are shown as "Cleaved RNA". DDX43 fails to unwind cleaved RNAs from Siwi-piRISC. − DDX43 means no DDX43 added in the reaction mixture.

the KH domain in DDX43 supports the ATP hydrolysis activity of the DEAD-box domain. The KH domain itself does not appear to be responsible for ATP hydrolysis, however, because the D399A and E400Q mutants also showed little ATP hydrolysis activity.

DDX43 WT and the D399A but not the E400Q mutant associated with Ago3 to a similar extent (Fig 4C). The ΔKH mutant also bound to Ago3 (Fig 4D). These results indicated that the KH domain is dispensable in the Ago3–DDX43 interaction but indispensable in ATP hydrolysis. The RNA-binding activity of DDX43 WT and the two mutants was verified by gel shift assays. DDX43 WT remained bound with RNA in the absence of ATP but dissociated from it in the presence of ATP (Fig 4E). In sharp contrast, the E400Q mutant remained bound to the RNA in the presence of ATP but did not bind to the RNA in the absence of ATP. The D399A mutant weakly bound the RNA under both conditions but the ΔKH mutant hardly bound to the RNA under any of the conditions we employed. These findings indicated that DDX43 WT can bind RNA in the absence of ATP but requires ATP for RNA dissociation, while the ΔKH mutant may not bind ATP because of a failure in RNA binding. The latter agreed with the above observation that this mutant exhibited little ATP hydrolysis activity. However, the ΔKH mutant bound with Ago3, indicating that the helicase core is dedicated to the DDX43–Ago3 interaction. The E400Q mutant was unique because it required ATP to bind the RNA, unlike DDX43 WT. One possible explanation for this is that the E400Q mutation resulted in a conformational change in

DDX43 and that DDX43–ATP binding reversed this change, allowing the mutant to bind RNA along with ATP.

We then mutated two amino acids in the VIGxxGxxI stretch conserved within the KH domains of many RNA-binding proteins (Siomi *et al*, 1994; Valverde *et al*, 2008), namely Gly90 and Ile96. These restudies were substituted with aspartic acid and asparagine, respectively, in the full-length protein and we then examined the ATP hydrolysis and RNA-binding activities of this double mutant (termed "GIDN"). This GIDN mutant hydrolyzed ATP as efficiently as the WT (Fig EV4B) but its RNA binding was negligible (Fig 4E), as has been observed for similar mutants of other KH domain-containing RNA-binding proteins (Siomi *et al*, 1994).

This series of investigations using DDX43 mutants clarified the roles of the two domains in DDX43 function. The helicase core is responsible for Ago3 binding and ATP hydrolysis, but hydrolysis requires physical (but not functional) support from the KH domain. The RNA-binding activity is cooperatively (but not additively) contributed by both domains.

Finally, we examined whether the D399A, ΔKH, and GIDN mutants of DDX43 exhibited the RNA releasing activity. All four mutants failed to show this activity in contrast to the WT control (Fig 4F and G). Thus, RNA release from Ago3-piRISC requires both the helicase core and the KH domain of DDX43.

Previous studies showed that human DDX43 exhibits RNA-unwinding activity *in vitro* and is expressed specifically in the germline, as well as in certain cancerous cells (Martelange *et al*, 2000;

**Figure 4. The KH domain in DDX43 is required for enzymatic activation.**

A   The domain organization of DDX43 and detail of the mutants. The conserved KH domain and DEAD-box helicase domain are indicated. D399 and E400 are highly conserved residues within the ATPase active site in the helicase domain.

B   Quantification of ATPase activity of DDX43 variants. The graph on the right indicates the ratio of the hydrolyzed ATP produced by DDX43 mutants to that of the WT DDX43. Data represent the mean ± standard deviations. $n = 3$ independent experiments. *$P < 0.002$ as determined with the *t*-test.

C   Immunoprecipitation of Ago3 with Flag-tagged DDX43 WT and its D399A and E400Q mutants. Ago3 and DDX43 were detected by Western blotting using anti-Ago3 and anti-Flag-antibodies, respectively. Flag-EGFP was used as a negative control. The E400Q mutation in DDX43 impairs the interaction between Ago3 and DDX43. IgG h.c.: IgG heavy chain.

D   Immunoprecipitation of Ago3 with Flag-tagged DDX43 WT and its ΔKH mutant lacking KH domain. Ago3 and DDX43 were detected by Western blotting using anti-Ago3 and anti-Flag-antibodies, respectively. Flag-EGFP was used as a negative control. The ΔKH mutant maintains affinity to Ago3. IgG h.c.: IgG heavy chain.

E   RNA-binding activity of DDX43 variants as analyzed by EMSA. Upper: The GIDN mutant was substituted G90 and I96 conserved within the KH domain to Asp and Asn, respectively. Lower: A 40-nt single-stranded RNA was 5′ end-labeled with $^{32}$P and incubated with DDX43 variants in the presence or absence of 0.1 mM ATP. Supershift means non-canonical complex.

F   RNA-unwinding assays upon Ago3-piRISC-dependent target RNA cleavage. Unwinding activities of DDX43 WT and its D399A, E400Q, and ΔKH mutants are shown. RNAs cleaved by Ago3 are shown as "Cleaved RNA".

G   RNA-unwinding assays upon Ago3-piRISC-dependent target RNA cleavage. Unwinding activities of DDX43 WT and its GIDN mutant are shown. RNA cleaved by Ago3 is shown as "3′ cleaved RNA".

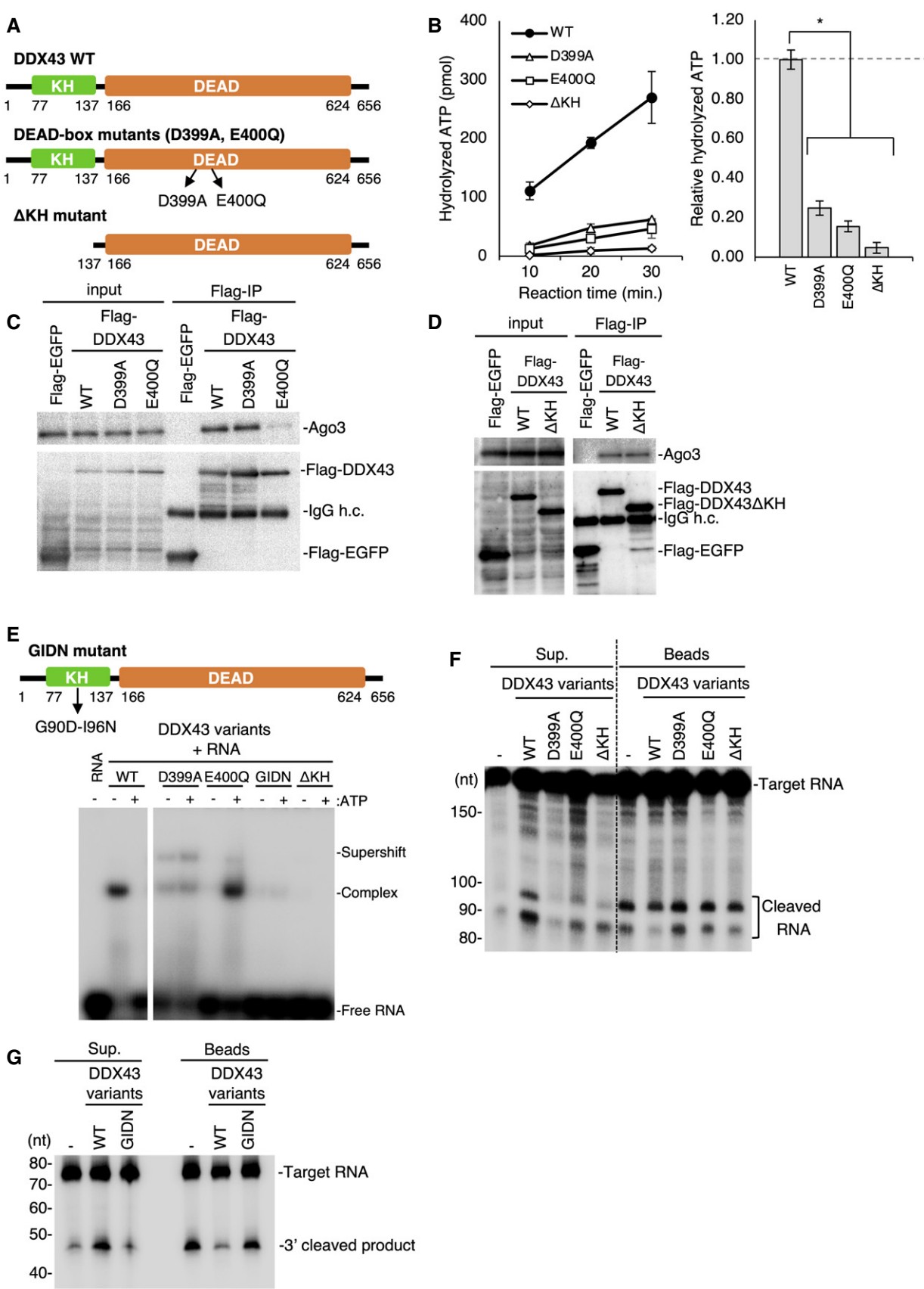

**Figure 4.**

Mathieu *et al*, 2010; Talwar *et al*, 2017). For this reason, DDX43 has been widely used as a germline marker, similar to Vasa (Gustafson & Wessel, 2010). However, the molecular function of this protein and the biological significance thereof remained unclear. In parallel, we and others previously proposed the function of Vasa in the ping-pong pathway (Xiol *et al*, 2014; Nishida *et al*, 2015): Vasa liberates RNA from Siwi-piRISC upon its cleavage to facilitate Ago3-piRISC production. However, Vasa failed to do the same for Ago3-piRISC to produce Siwi-piRISC. This meant that there was another Vasa-like factor acting on Ago3-piRISC to remove RNAs from the complex for Siwi-piRISC synthesis. However, this factor remained unidentified since the discovery of Vasa function in the piRNA pathway in 2015. In the current study, we succeeded in identifying a "sibling" of Vasa, DDX43, as the missing factor and provided evidence that DDX43 indeed acts as the key player in Siwi-piRISC synthesis through the ping-pong cycle.

How DDX43 mechanistically liberates cleaved RNAs from Ago3-piRISC remains unknown. The RNA-unwinding activity of human DDX43 shows a strong bias toward substrates containing a 5′-overhang (Talwar *et al*, 2017). If this bias also applies to *Bombyx* DDX43, the 5′-fragment of the two RNA fragments arising from Ago3-dependent cleavage should theoretically be released more quickly from the piRISC than the 3′-fragment. However, in reality, we observed that the 3′-fragment containing a 3′-overhang was released from the piRISC faster than the 5′-fragment containing a 5′-overhang. This suggests that the direction of RNA unwinding by *Bombyx* DDX43 is opposite to that of human DDX43. The 3′-fragment anneals with Ago3-bound piRNA over a 10-nt stretch, while the 5′-fragment anneals over 16 nt or more (depending on the length of the piRNA). From a thermodynamic point of view, the 5′-fragment is, therefore, more stably bound to the piRNA than the 3′-fragment, which would explain why this fragment is released more slowly. What if a target RNA were used harboring a sequence complementary to the piRNA over only 20 nt from the 5′-end of the piRNA? This would equal the number of nucleotides in the piRNA annealing to both the 5′- and 3′-fragments combined (i.e., 10 nt each), and thermodynamic bias would no longer be evident. However, in this case, Ago3 cleavage may not be effective and the result would be inconclusive because such treatment would destabilize the target RNA–piRNA interaction. An alternative hypothesis is that DDX43 liberates cleaved RNAs from the piRISC from the site where Ago3-dependent cleavage occurs.

DDX43 contains the KH domain in addition to the helicase core. To our knowledge, this domain combination is unique to DDX43 among the members of the DEAD-box protein family. The KH domain of DDX43 exhibits RNA-binding activity, as expected. Interestingly, however, this domain was observed to play an additional role, namely the acceleration of the ATP hydrolysis activity of the helicase core, although this enhancement of ATP hydrolysis was not dependent on the RNA-binding activity of the KH domain. This inter-domain regulation of protein functions is, however, not exclusive to DDX43. RNase R is an exonuclease with helicase activity and consists of a nuclease domain, cold-shock domains (CSD1, 2), and S1 domain (Vincent & Deutscher, 2009; Hossain *et al*, 2015; Hossain *et al*, 2016). The CSDs and S1 additively exert the double-stranded RNA-binding and ATP-binding activity of RNase R, which then strengthens its unwinding activity. Prp22, a DExD/H-box protein playing a role in splicing, is composed of an S1 domain and a helicase core (Schneider & Schwert, 2001). Interestingly, the S1 domain has a negative effect on the ATPase activity, as well as the RNA-unwinding activity of the helicase core of Prp22.

Vasa and DDX43 release RNAs from Siwi-piRISC and Ago3-piRISC, respectively. How is this selectivity maintained *in vivo*? Domain mapping may help us to answer this question. Alternatively, we postulate that solving the 3D structures of Siwi-piRISC bound with Vasa and Ago3-piRISC bound with DDX43 in the absence of $Mg^{2+}$ may be a more straightforward approach. However, this may require the piRISC to be in a form that is engaged with the target RNAs. We previously solved the 3D structure of endogenous Siwi-piRISC (Matsumoto *et al*, 2016). At that time, the piRISC was immunoisolated from BmN4 cells using an anti-Siwi antibody, and the piRNAs residing in the structure were heterogeneous in sequence. To investigate the piRISCs in a target RNA-engaged form, we would need to use complexes containing only one piRNA sequence at a time. Because isolating unbound PIWI is a challenging task (due to its extremely high instability *in vivo*), we are currently developing a method for substituting the originally isolated piRNAs with a new piRNA of choice.

# Materials and Methods

### Plasmid construction

The primers used to produce the constructs are listed in Table EV1. Vectors to express Flag-EGFP, Flag-Siwi, Flag-Ago3, Flag-Spn-E, and Flag-Vasa were constructed as described previously (Nishida *et al*, 2015). To express Flag-tagged Dpb-2, DDX43, Bel, and me31B in BmN4 cells, these genes were amplified by PCR using BmN4 cDNA and then cloned into pIB-3xFlag vector using NEBuilder HiFi DNA Assembly Master Mix (New England Biolabs, Ipswich, MA, USA) or SLiCE (Motohashi, 2015). To yield the pET47b-DDX43 and pGEX6p1-DDX43 N-200 (amino acids 1-200), DNA fragments of DDX43 were amplified by PCR using pIB-3xFlag-DDX43 and inserted into pET-47b or pGEX-6p1. The expression vectors of the DDX43 mutants [G90D-I96N (GIDN), D399A, E400Q, and ΔKH)] and the EGFP mRNA inserted Ago3 target sequence were prepared by site-directed mutagenesis.

### RNAi and transgene expression in BmN4 cells

Transfection of the siRNA and expression vector were performed essentially as described previously (Nishida *et al*, 2015). The sequences of the siRNA duplexes used for RNAi are listed in Table EV1. To knock down genes, BmN4 cells ($0.5–1 \times 10^6$ cells) were transfected with 500 pmol siRNA duplex by electroporation with the Nucleofector 2b (Lonza, Basel, Swiss). To exogenously express proteins in BmN4 cells, the cells ($6 \times 10^5$ cells) were transfected with 2 μg of plasmid in 5 μl of FuGENE HD transfection reagent (Promega, Madison, WI, USA). After transfection, cells were incubated at 26°C for 48 h.

### Antibodies

Monoclonal antibodies for Vasa, Siwi, and Ago3 detection were produced as described previously (Nishida *et al*, 2015). The anti-

Vret monoclonal antibody was raised in our laboratory (Nishida *et al*, 2020). We produced the anti-DDX43 monoclonal antibody by immunizing mice with the purified GST-tagged N-terminal region (amino acids 1–200) of DDX43. The monoclonal ANTI-FLAG M2 antibody (Merck, Darmstadt, Germany) for immunoprecipitation, anti-DDDDK-tag monoclonal antibody (FLA-1 clone, Medical and Biological Laboratories, Aichi, Japan) for Western blotting and non-immune mouse IgG (Immuno-Biological Laboratories, Gunma, Japan) were commercially sourced.

## Immunoprecipitation

Immunoprecipitation was performed essentially as described previously (Saito *et al*, 2006). For co-immunoprecipitation, BmN4 lysates prepared in Co-IP buffer [30 mM HEPES-KOH pH 7.3, 150 mM potassium acetate, 5 mM magnesium acetate, 5 mM dithiothreitol, 0.1% (*w/v*) NP-40 and cOmplete ULTRA EDTA-free protease inhibitor (Roche, Indianapolis, IN, USA)] were incubated with antibodies bound to Dynabeads Protein G (Thermo Fisher Scientific, Waltham, MA, USA) at 4°C for 1 h. The beads were washed three times with Co-IP buffer, and then isolated proteins were eluted with SDS sample buffer for Western blotting or silver staining using the SilverQuest Silver Staining Kit (Thermo Fisher Scientific). For immunoprecipitation in the presence of EDTA, we used Co-IP buffer containing 1 mM EDTA instead of magnesium acetate. To avoid co-elution of the antibody during the identification of proteins associated with Ago3, the beads and antibody were crosslinked with phosphate-buffered saline (PBS) containing 200 mM triethanolamine and 50 mM dimethyl pimelimidate at 26°C for 1 h. After incubation, the reaction was quenched with PBS containing 50 mM ethanoleamine at 26°C for 15 min, and the beads were then incubated with 0.1 M glycine pH 3.0 at 26°C for 10 min to remove uncrosslinked antibody. The crosslinked beads were washed with 0.1 M glycine (pH 3.0) and equilibrated with Co-IP buffer before immunoprecipitation. For the preparation of beads bound only with Siwi or Ago3, we performed immunoprecipitation using HS buffer (Co-IP buffer containing 500 mM sodium chloride) instead of Co-IP buffer.

## Purification of recombinant proteins

The pIB vectors expressing Flag-tagged DDX43 variants or Vasa were transfected into BmN4 cells. The cells were suspended with HS buffer and incubated for 10 min on ice. After incubation, samples were centrifuged to remove the cell debris and applied to ANTI-FLAG M2 Affinity Gel (Merck) equilibrated with HS buffer. The resin was then rotated at 4°C for 1 h and then washed with five column volumes of HS buffer and three column volumes of Co-IP buffer. The immunoisolated proteins were eluted with Co-IP buffer containing 10% (*v/v*) glycerol and 500 μg/ml 3× FLAG Peptide (Merck). pET and pGEX vectors encoding His-tagged or GST-tagged proteins, respectively, were transformed to Rosetta2 (DE3) cells (Merck). The cells were grown at 37°C in Luria Bertani medium until the optical density at 600 nm reached 0.6, following which, protein expression was induced by the addition of 0.2 mM isopropyl β-D-1-thiogalactopyranoside (IPTG) at 16°C for 18 h. To purify the His-tagged DDX43 variants, the cells were suspended with His A buffer [20 mM Tris–HCl pH 8.0, 1 M sodium chloride, 20 mM imidazole, 1 mM dithiothreitol, 5% (*v/v*) glycerol and cOmplete

ULTRA EDTA-free protease inhibitor (Roche)] and lysed by sonication. The samples were centrifuged and filtered to remove the cell debris, and then loaded onto Ni sepharose 6 Fast Flow resin (GE Healthcare, Chicago, IL, USA) equilibrated with His A buffer. The resin was washed with 10 column volumes of His A buffer, and bound proteins were eluted with His B buffer [20 mM Tris–HCl pH 8.0, 1 M sodium chloride, 300 mM imidazole, 1 mM dithiothreitol and 5% (*v/v*) glycerol]. The eluted proteins were dialyzed against storage buffer [25 mM HEPES-KOH pH 7.3, 150 mM potassium acetate, 1 mM dithiothreitol and 20% (*v/v*) glycerol]. To yield GST-DDX43 N-200, the cells were suspended with GSTA buffer [20 mM Tris–HCl pH 8.0, 500 mM sodium chloride, 1 mM dithiothreitol, and cOmplete ULTRA EDTA-free protease inhibitor (Roche)] and lysed by sonication. The sample was clarified by centrifugation and filtration and loaded onto Glutathione Sepharose 4B resin (GE Healthcare). The resin was washed with 10 column volumes of GSTA buffer, and bound protein was eluted with GSTB buffer (50 mM Tris–HCl pH 8.0, 500 mM sodium chloride, 1 mM dithiothreitol and 10 mM reduced glutathione). After elution, the sample was dialyzed against PBS containing 10% (*v/v*) glycerol. After dialysis, the purified proteins were concentrated with an Amicon Ultra-4 centrifugal filter unit (Merck) and quantified with the Pierce BCA Protein Assay Kit (Thermo Fisher Scientific).

## Isolation and detection of small RNAs associated with PIWI proteins

Small RNAs associated with Siwi and Ago3 were extracted from immunoprecipitated beads by phenol-chloroform treatment and precipitated with ethanol. RNAs were dephosphorylated with antarctic phosphatase (New England Biolabs), radiolabeled with gamma-$^{32}$P-ATP using T4 Polynucleotide Kinase (New England Biolabs) and then separated on a denaturing PAGE. Autoradiographs were captured with the using Typhoon FLA 9500 laser scanner (GE Healthcare).

## Northern blotting

Northern blotting and isolation of total RNA from BmN4 cells were performed essentially as described previously (Hirakata *et al*, 2019). The sequences of the probes used are shown in Table EV1. Autoradiographs were captured with the Typhoon FLA 9500 scanner, and the signal intensities were calculated using ImageJ software (https://imagej.nih.gov/ij/).

## Mass spectrometry

The solutions containing peptides were applied to liquid chromatography–tandem mass spectrometry (LC-MS/MS) analysis whereas the analytical details have already been reported (Yashiro *et al*, 2018). NanoLC-MS/MS analysis was conducted by LTQ-Orbitrap Velos mass spectrometer (Thermo Fisher Scientific) equipped with Zaplous Advance nano UHPLC HTS-PAL xt System (AMR). The mass spectrometer was operated in the positive ionization mode, and isolated charged ions were sequentially fragmented in the linear ion trap by collision-induced. The protein annotation data were compared against the protein sequence data available for *Bombyx mori* in the UniProt database through the application of the search

program Proteome Discoverer 2.1 (Thermo Fisher Scientific) featuring the Sequest HT search algorithm for the identification of proteins and the post-translational modification sites. The carbamidomethylation of Cys was searched as a fixed modification, whereas oxidized Met, phosphorylation of Ser, Thr, and Tyr were searched as variable modifications.

**Pull-down assay**

The beads bound with Ago3 were prepared as described earlier in the Materials and Methods and mixed with 200 µl Co-IP buffer containing 5 ng/µl His-DDX43. The mixture was rotated at 4°C for 1 h and washed twice with Co-IP buffer. The bound proteins were eluted with SDS sample buffer and analyzed by SDS–PAGE, followed by silver staining.

**Immunofluorescence**

BmN4 cells were placed on 0.075% ($w/v$) poly-L-lysine coated cover glasses. Cells were fixed with 4% formaldehyde in PBS for 15 min and permeabilized with 0.2% ($v/v$) Triton X-100 in PBS for 15 min. After blocking with 3% ($w/v$) BSA in PBS (PBS-B) for 15 min, cells were incubated with 1 ng/µl primary antibodies diluted with PBS-B for 1 h. After washing with PBS-B, cells were incubated with secondary antibodies [Alexa 488- and Alexa Fluor 555-labeled goat anti-mouse immunoglobulin G1 (IgG1) and IgG2a antibodies (Thermo Fisher Scientific)] in PBS-B for 1 h in a dark room. Cover glasses were mounted with Vectashield Antifade Mounting Medium with DAPI (Vector Laboratories, Burlingame, CA, USA). Images were captured with an LSM 980 confocal laser-scanning microscope (Zeiss) equipped with Plan-Apochromat 63×/1.4 Oil DIC M27 objective lens (Zeiss).

**Preparation of target RNAs**

In Figs 3D, E, 4F, and EV3D, the template DNA for the *in vitro* transcription of the Ago3 target RNA was amplified by PCR using primers for the T3 and T7 promoter. The radiolabeled target RNA was transcribed with the MEGAscript T3 Transcription Kit (Thermo Fisher Scientific) and labeled with α-$^{32}$P-UTP. After transcription, the RNA sample was purified by denaturing PAGE. In Fig 3F, Siwi target RNA was radiolabeled using T4 polynucleotide kinase. In Fig 4G, to produce the target RNA, Ago3 target RNA 5′ and radiolabeled Ago3 target RNA 3′ were ligated using T4 RNA Ligase 2 (New England Biolabs) and purified by denaturing PAGE. The sequences of the template DNA used for *in vitro* transcription and RNAs were described in Table EV1.

**In vitro RNA cleavage and releasing assay**

The magnetic beads (600 µg) bound with Ago3 were prepared as described earlier in the Materials and Methods and mixed with 100 µl cleavage buffer [20 mM HEPES-KOH pH 7.3, 100 mM potassium acetate, 2 mM magnesium acetate, 5 mM dithiothreitol, and 0.04 U/µl RNasin Ribonuclease inhibitors (Promega)] containing 5 µg/ml yeast RNA (Thermo Fisher Scientific) and $^{32}$P-labeled Ago3 target RNA. After incubation for 3 h at 26°C, the reaction mixture was separated into the supernatant and the beads. For

cleaved RNA releasing assay, the separated beads were mixed with 100 µl cleavage buffer containing 4 mM ATP and 130 nM Flag-Vasa or Flag-DDX43, incubated at 26°C for 1 h, and then, the mixtures were separated into the beads and the supernatant. The target RNAs in these fractions were extracted by phenol-chloroform treatment, precipitated with ethanol, run on denaturing gels, and visualized with the Typhoon FLA 9500. The assays using Siwi were performed based on the method described previously (Nishida *et al*, 2015). The magnetic beads (900 µg) bound with Siwi were prepared as described earlier in the Materials and Methods and mixed with 30 µl Siwi cleavage buffer [50 mM Tris–HCl pH 8.0, 150 mM sodium chloride, 2 mM magnesium chloride, 1 mM dithiothreitol, 5% ($w/w$) PEG6000, 0.04 U/µl RNasin Ribonuclease inhibitors (Promega) and 5 µg/ml yeast RNA (Thermo Fisher Scientific)] and $^{32}$P-labeled Siwi target RNA. After incubation for 3 h at 26°C, the reaction mixture was separated into the supernatant and the beads. After separation, the beads were mixed with 30 µl Siwi cleavage buffer containing 4 mM ATP, 10 mM creatine phosphate, 30 µg/ml creatine kinase (Roche), and 130 nM Flag-Vasa or Flag-DDX43, incubated at 26°C for 2 h. The mixtures were then separated into the beads and the supernatant. Target RNAs in these fractions were extracted and detected as in Ago3 cleavage assays.

**In vivo reporter assay**

BmN4 cells ($5 \times 10^6$ cells) were transfected with luciferase, Ago3, or DDX43 siRNA duplex as described earlier in the Materials and Methods. After 6 days, the cells ($5 \times 10^6$ cells) were transfected again with the same siRNA duplex. The cells were transfected with 10 µg reporter plasmid (pIB-EGFP-Ago3 target) and pIB-Flag-Siwi on day 7 and 8, respectively. On day 10, the total RNA and Flag-Siwi-piRNA were isolated from the cells and analyzed as described earlier in the Materials and Methods.

**Electrophoretic mobility shift assay (EMSA)**

For the gel shift assay, 40 nt RNA was radiolabeled with γ-$^{32}$P-ATP using T4 polynucleotide kinase and purified on Micro Bio-Spin Columns loaded with Bio-Gel P-30 (Bio-Rad, Hercules, CA, USA). The 10-µl reaction mixtures containing 25 mM HEPES-KOH (pH 7.3), 50 mM potassium acetate, 1 mM magnesium acetate, 1 mM dithiothreitol, 5 nM the labeled RNA, and 0.5 µM His-DDX43 variants were incubated at 26°C for 10 min. After incubation, the samples were mixed with 2 µl of nondenaturing PAGE sample buffer [0.1% ($w/v$) bromophenol blue, 0.1% ($w/v$) xylene cyanol, and 50% ($v/v$) glycerol] and then the RNA-Protein complexes were separated on a 6% nondenaturing Tris-glycine gel. The gels were visualized with the Typhoon FLA 9500.

**Measurement of ATPase activity**

The 20-µl reaction mixtures, containing 0.5 µM DDX43, 25 mM HEPES-KOH (pH 7.3), 50 mM potassium acetate, 1 mM magnesium acetate, 1 mM dithiothreitol, 0.5 mg/ml BSA and 2 mM gamma-$^{32}$P-ATP, were incubated at 26°C. After incubation, the inorganic phosphate liberated was assayed as described previously (Frolova *et al*, 1996).

## Analysis of RNA sequences

RNA sequencing libraries were prepared using the NEBNext Multiplex Small RNA Library Prep Set for Illumina (New England BioLabs) and then sequenced with the Illumina MiSeq (single-end, 51 cycles). After adaptor sequences were removed from obtained reads, for Siwi-piRNAs, total reads were obtained from DDX43-knockdown samples (library-1 and library-2, respectively). Siwi-piRNA libraries under control and Vasa-knockdown conditions were used as described in a previous study (Nishida *et al*, 2015). The reads were also mapped to the *B. mori* reference genome downloaded from Silk-Base (silkbase.ab.a.u-tokyo.ac.jp) using Bowtie software (http://bowtie-bio.sourceforge.net), with no mismatch allowed (Langmead *et al*, 2009). Using sequences mapped to the genome, phasing analyses were conducted. For Fig EV2A, the reads in library-1 and library-2 from Siwi-piRNA in the DDX43-knockdown samples described above were normalized to reads per million (RPM) against the total number of reads in the corresponding dataset. After extracting the reads with scores > 10 RPM, Spearman's correlation was calculated and scatter plot was described for each sample. For Fig EV2B, the RPM scores of library-1 and library-2 were first combined for each sample. Comparing the control and Vasa-knockdown samples, the reads were classified into three groups based on their Vasa-KD (RPM)/Control (RPM) ratio: piRNAs with a ratio Vasa-KD (RPM)/Control (RPM) value was $\geq 2$, $0.5 < \text{ratio} < 2$, or ratio $\leq 0.5$ were categorized as increased, unchanged, or decreased, respectively.

## Sequence alignment

The multiple sequence alignment was generated with the programs Clustal W (Larkin *et al*, 2007) and ESPript (Robert & Gouet, 2014).

# Data availability

The accession number for the sequence data reported in this paper is as follows GEO: GSE161405. The mass spectrometry data from this publication have been deposited to the ProteomeXchange (http://www.proteomexchange.org) and assigned the identifier PXD023467.

**Expanded View** for this article is available online.

## Acknowledgements

We thank K.M. Nishida and H. Yamada for technical suggestions, reading manuscript and useful comments. We also thank A. Takemura for technical support. We also thank S. Uemura and all other members of the Siomi laboratory at the University of Tokyo. This study was supported by research grants from MEXT to M.C.S. (19H05466), JST-CREST (JPMJCR14W1) to M.C.S., Grant-in-Aid for Early-Career Scientists to R.M. (18K14621), and The Sumitomo Foundation Fiscal 2020 grant for basic science research projects to R.M (180164).

## Author contributions

RM performed biochemical analyses. TS performed bioinformatics analyses. LN performed LC–MS/MS analyses. RM and MCS designed the experiments and wrote the manuscript. MCS supervised all of the research.

## Conflict of interest

The authors declare that they have no conflict of interest.

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
