## [Review Process File · EMBO Reports]

DEAD-box polypeptide 43 facilitates piRNA amplification by actively liberating RNA from Ago3-piRISC

Ryo Murakami, Tetsutaro Sumiyoshi, Lumi Negishi, and Mikiko Siomi
DOI: [10.15252/embr.202051313](https://doi.org/10.15252/embr.202051313)

Corresponding author(s): Mikiko Siomi (siomim@bs.s.u-tokyo.ac.jp)

Review Timeline:

Submission Date:	13th Jul 20
Editorial Decision:	20th Aug 20
Revision Received:	18th Nov 20
Editorial Decision:	16th Dec 20
Revision Received:	28th Dec 20
Accepted:	8th Jan 21

Editor: Esther Schnapp & Achim Breiling

Transaction Report:

Dear Prof. Siomi,

Thank you for the submission of your manuscript to EMBO reports. We have now received reports from the two referees that were asked to evaluate your study, which can be found at the end of this email.

As you will see, both referees think that the findings are of high interest, but they also have several comments and suggestions, indicating that a major revision of the manuscript is necessary to allow publication in EMBO reports. As the reports are below, and I think all points need to be addressed, I will not detail them here.

Given the constructive referee comments, we would like to invite you to revise your manuscript with the understanding that all referee concerns must be addressed in the revised manuscript and/or in a detailed point-by-point response. Acceptance of your manuscript will depend on a positive outcome of a second round of review. It is EMBO reports policy to allow a single round of revision only and acceptance of the manuscript will therefore depend on the completeness of your responses included in the next, final version of the manuscript.

Revised manuscripts should be submitted within three months of a request for revision. We are aware that many laboratories cannot function at full efficiency during the current COVID-19/SARS-CoV-2 pandemic and we have therefore extended our 'scooping protection policy' to cover the period required for full revision. Please contact me to discuss the revision should you need additional time, and also if you see a paper with related content published elsewhere.

- 1) a .docx formatted version of the final manuscript text (including legends for main figures, EV figures and tables), but without the figures included. Please make sure that changes are highlighted to be clearly visible. Figure legends should be compiled at the end of the manuscript text.
- 2) individual production quality figure files as .eps, .tif, .jpg (one file per figure), of main figures and EV figures. Please upload these as separate, individual files upon re-submission.

The Expanded View format, which will be displayed in the main HTML of the paper in a collapsible format, has replaced the Supplementary information. You can submit up to 5 images as Expanded View. Please follow the nomenclature Figure EV1, Figure EV2 etc. The figure legend for these should be included in the main manuscript document file in a section called Expanded View Figure Legends after the main Figure Legends section. Additional Supplementary material should be supplied as a single pdf file labeled Appendix. The Appendix should have page numbers and needs

to include a table of content on the first page (with page numbers) and legends for all content. Please follow the nomenclature Appendix Figure Sx, Appendix Table Sx etc. throughout the text, and also label the figures and tables according to this nomenclature.

For more details please refer to our guide to authors:

See also our guide for figure preparation:

http://wol-prod-cdn.literatumonline.com/pb-assets/embosite/EMBOPress_Figure_Guidelines_061115-1561436025777.pdf

4) a complete author checklist, which you can download from our author guidelines (<https://www.embopress.org/page/journal/14693178/authorguide>). Please insert page numbers in the checklist to indicate where the requested information can be found in the manuscript. The completed author checklist will also be part of the RPF.

Please also follow our guidelines for the use of living organisms, and the respective reporting guidelines: <http://www.embopress.org/page/journal/14693178/authorguide#livingorganisms>

5) that primary datasets produced in this study (e.g. RNA-seq, ChIP-seq and array data) are deposited in an appropriate public database. This is now mandatory (like the COI statement). If no primary datasets have been deposited in any database, please state this in this section (e.g. 'No primary datasets have been generated and deposited').

The accession numbers and database should be listed in a formal "Data Availability " section (placed after Materials & Methods) that follows the model below. Please note that the Data Availability Section is restricted to new primary data that are part of this study.

Data availability

6) We strongly encourage the publication of original source data with the aim of making primary data more accessible and transparent to the reader. The source data will be published in a separate source data file online along with the accepted manuscript and will be linked to the relevant figure. If you would like to use this opportunity, please submit the source data (for example scans of entire gels or blots, data points of graphs in an excel sheet, additional images, etc.) of your key experiments together with the revised manuscript. If you want to provide source data, please include size markers for scans of entire gels, label the scans with figure and panel number, and send one PDF file per figure.

8) Regarding data quantification and statistics, can you please specify, where applicable, the number "n" for how many independent experiments (biological replicates) were performed, the bars and error bars (e.g. SEM, SD) and the test used to calculate p-values in the respective figure legends. Please provide statistical testing where applicable, and also add a paragraph detailing this to the methods section. See: <http://www.embopress.org/page/journal/14693178/authorguide#statisticalanalysis>

9) Please also note our new reference format:
<http://www.embopress.org/page/journal/14693178/authorguide#referencesformat>

I look forward to seeing a revised version of your manuscript when it is ready. Please let us know if you have questions or comments regarding the revision.

Kind regards,

Achim

Achim Breiling
Editor
EMBO Reports

Referee #1:

The manuscript by Murakami et al., investigates the role of an RNA helicase in the piRNA pathway. The piRNA pathway is exclusively expressed in animal gonads and they repress transposable elements. Apart from the gonads itself, a few cell culture models are available for research. One widely used model system is the Silkworm ovarian cell line BmN4.

The piRNA pathway in BmN4 cells consists of two Piwi proteins Siwi and Ago3. Biogenesis of

piRNAs in this system depends on the Ping-pong cycle or piRNA amplification. This involves targeting an RNA by a piRNA-loaded Piwi protein, endonucleolytic cleavage of the target into two fragments, and loading of one of the fragments into the opposite Ping-pong partner. How this fragment is transferred from one Piwi protein to the other is a very important question, as it has to be specific and largely heterotypic (Siwi to Ago3 or Ago3 to Siwi). Previous studies (including from the authors) have implicated the RNA DEAD box helicase Vasa in transferring the cleaved fragment from Siwi to Ago3. Now the authors identify another RNA DEAD box helicase called DDX43 and show that it helps in Ago3 to Siwi transfer.

The results are novel and convincing. They point an RNA helicase that specifically acts on Ago3-piRISC to release the cleavage fragment for loading into Siwi. They demonstrate that the helicase core domain is involved in this interaction. This study will be of immense value for the small RNA community.

Minor:

1. Page 4: "3'-end fragment is loaded onto piRNA-unbound Ago3, giving rise to the Ago3-piRISC precursor (pre-Ago3-piRISC). The 5'-end of the RNA fragment" The use of the term "3' end fragment" and in the next sentence "5'-end of the RNA fragment" is confusing. This is used throughout the text. "Fragment with a 5' phosphate?"
2. Fig2D. Interaction between DDX43 and Ago3 in the absence of Vret need some explanation. Why would Ago3-piRISC that is not engaged with a target be a good interaction partner for DDX43, compared to that engaged with a target, which is the logical substrate for DDX43?
3. Fig3E. Why is there more bottom cleaved RNA fragment in the Sup after treatment with DDX43. Lane numbers might be useful.
4. Is interaction between DDX43 and Ago3 mediated by sDMA modifications?

Referee #2:

Review of manuscript "DEAD-box polypeptide 43 facilitates piRNA amplification by actively liberating RNA from Ago3-piRISC" by Mikiko Siomi and colleagues.

In this manuscript, Siomi and colleagues build on their previous interesting observation that the ping-pong partner Argonautes Ago3 and Aubergine/Siwi remain associated with their targets after target cleavage. In thinking about this, such a scenario makes immediate sense as the cleavage products of Ago3 and Aubergine/Siwi are not meant to be degraded, but instead one of the cleavage products (the one with slicer-generated 5' phosphate) will give rise to the new ping-pong partner piRNA. By holding on to their cleavage products, Ago3 and Aubergine/Siwi thereby protect the cleavage products until they are properly funneled into the partner protein. In their previous work, Siomi and colleagues made the important discovery that the RNA helicase Vasa, a central piRNA biogenesis factor is required to release the cleaved target products from Aubergine/Siwi. Consistent with this, depletion of Vasa has a profound impact on piRNA biogenesis and Ago3-bound piRNAs are eliminated, while Aubergine/Siwi bound piRNAs remain abundant (yet shift from ping-pong piRNAs to so-called primary piRNAs, which are considered ping-pong independent). The authors also found that Vasa is not able to release the cleavage products from Ago3, and therefore they postulated that a Vasa-analogous factor should/might exist for Ago3.

In this current work, the authors present data that indicates that the RNA helicase DDX43 is the sought-for protein that releases RNA cleavage products from Ago3. This would be a very

worthwhile and important finding. DDX43 is so far not linked to the piRNA pathway, and if true, this finding would answer an obvious question in the piRNA biogenesis field. As such, the findings would be very well suited for EMBO reports.

All in all, I would like to congratulate the authors on this work, which presents multiple lines of evidence that converge on a role of DDX43 in the postulated process. I have, however, a few comments that in my opinion must be adequately resolved before I can recommend publication of this work. My main comments all circle around the central question, whether it is possible that the presented data (despite being technically overall well controlled) represent a false positive. Meaning that yes, DDX43 in vitro has the reported ability to release cleavage products from Ago3, but in vivo it has a different function.

My main three questions/concerns:

1. Where does DDX43 localize in vivo? It is surprising that this rather obvious question is not touched upon in this work. As the authors point out in the text related to Figure 2, piRNA biogenesis is 'enriched' in peri-nuclear structures, generally referred to as nuage. Indeed, many piRNA biogenesis factors (especially those involved in ping-pong) are enriched in nuage (some other factors are found on the mitochondrial surface where 3' end processing of many piRNAs as well as phased piRNA biogenesis occurs). Vasa for example is a prototypical nuage marker, consistent with its role in releasing the cleavage products from Aubergine/Siwi. It would therefore be very much expected that DDX43 also localizes to nuage. Moreover, nuage localization of DDX43 to nuage should be Ago3 dependent according to the author's model.

It would be important to carefully determine and evaluate the subcellular localization of DDX43 in BmN4 cells and to bring these findings into context with the other findings of this paper. If, for example, DDX43 would localize to the nucleus, this would argue against a role of DDX43 in Ago3 biology in vivo.

2. A key concern in my opinion is that the in vivo data supporting a role of DDX43 in Ago3 biology is on the weak side. The authors use RNAi to deplete DDX43 in BmN4 cells, and they find pretty much no measurable phenotype when analyzing piRNA populations bound to either Ago3 or Aubergine/Siwi. Given the strong statements in this paper that DDX43 is the factor that is required for Ago3 cleavage product release, this is certainly surprising. The authors mention that this might be caused by incomplete knockdown. That is per se possible, yet not very satisfying. Given that Cas9-mediated knockouts are in principle possible in BmN4 cells (see Tomari lab for example), I strongly suggest that a certified loss of function analysis is required to support the in vivo relevance of DDX43 in the postulated process. I would like to point out that the experiment presented in Figure 3B,C is an interesting way to show that DDX43 might indeed play a role in release of Ago3 cleavage products (and hence in piRNA biogenesis fueled into Aub). It is, however, a single case experiment and given this result it is surprising that genome wide not even a slight trend in defective Siwi-piRISC formation (ping-pong centered) can be observed (Figure EV2). An alternative approach could be to transition to the Drosophila system where highly potent ways to deplete target proteins via transgenic RNAi exist, and to test here for a (presumably) conserved function of DDX43 in the piRNA ping-pong cycle. I appreciate that this point overall involves quite a bit of effort, yet in my opinion demonstrating a clear in vivo relevance for this protein in the ping-pong cycle would be required.

3. My third point relates to the question whether DDX43 is specific for the cleavage product release from Ago3. The authors mention in their introduction that DDX43 acts on Ago3, yet not on Siwi. I might have overlooked this data, but I cannot find in the entire manuscript a datapoint that shows that addition of DDX43 to Siwi engaged with cleaved targets would not release these. I consider

demonstrating that DDX43 is indeed specific to Ago3 as highly relevant. The authors show that such a specificity is indeed the case for Vasa. It only releases cleavage products from Siwi, but not from Ago3. Such a targeted experiment is required for DDX43 as well in my opinion. If DDX43 is also releasing Siwi's cleavage products, the in vitro ability of this RNA helicase to remove target fragments from an Argonaute might be true in vitro, but not a reality in vivo.

One additional aspect that I consider relevant is to compare DDX43 mutant conditions to Ago3 mutant conditions in terms of RNA-seq as well as piRNA sequencing. How much are Siwi-bound ping-pong piRNAs affected in an Ago3 depleted cell? And if they are strongly affected, would this not speak against a role of DDX43 as an essential release factor for Ago3 given that no changes in ping-pong piRNA levels are seen in DDX43 depleted cells? Moreover, I would consider it important to exclude (or at least to ask for) a potential role of DDX43 in non ping-pong processes. A simple RNAseq experiment on Ago3-depleted versus DDX43 depleted cells should give a good indication whether loss of DDX43 leads to many mis-regulated genes. If that were the case, the authors ideally acknowledge this and build this finding into their discussion.

Minor comments:

- Figure 1E: the vast majority of Ago3 in this experiment is expected to be mature Ago3-RISC, not associated with a target. Why would then DDX43 bind nearly stoichiometrically? Is this interaction also dependent on ATP and is it weaker in Mg²⁺ containing buffers?
- How does the target release kinetics compare between Ago3 and Siwi (DDX43 versus Vasa)?

Referee #1:

The results are novel and convincing. They point an RNA helicase that specifically acts on Ago3-piRISC to release the cleavage fragment for loading into Siwi. They demonstrate that the helicase core domain is involved in this interaction. This study will be of immense value for the small RNA community.

We thank the referee for these positive comments.

Minor:

1. Page 4: "3'-end fragment is loaded onto piRNA-unbound Ago3, giving rise to the Ago3-piRISC precursor (pre-Ago3-piRISC). The 5'-end of the RNA fragment" The use of the term "3' end fragment" and in the next sentence "5'-end of the RNA fragment" is confusing. This is used throughout the text. "Fragment with a 5' phosphate?"

We thank the referee for pointing this out. To address this issue, we now refer to "3'-end fragment" as "3'-fragment" throughout the manuscript.

2. Fig2D. Interaction between DDX43 and Ago3 in the absence of Vret need some explanation. Why would Ago3-piRISC that is not engaged with a target be a good interaction partner for DDX43, compared to that engaged with a target, which is the logical substrate for DDX43?

In this study, we provide evidence for the claim that DDX43 is the factor liberating cleaved RNAs from Ago3-piRISC to facilitate Siwi-piRISC production in the ping-pong cycle. This is similar to Vasa liberating cleaved RNAs from Siwi-piRISC to facilitate Ago3-piRISC production along the same pathway. We previously showed that Vasa binds Siwi-piRISC in a target RNA-independent manner (Nishida *et al.*, 2015). Based on this finding, it is not surprising that DDX43 binds Ago3-piRISC in a target RNA-independent manner.

3. Fig3E. Why is there more bottom cleaved RNA fragment in the Sup after treatment with DDX43. Lane numbers might be useful.

As we had already noted in the original manuscript (page 13), the 3'-fragment (*i.e.*, the shorter one close to the gel front; 83 nt) base-pairs with Ago3-bound piRNA over a 10-nt stretch, while the 5'-fragment (*i.e.*, the longer one; 91 nt) base-pairs over 16-nt, which is a more stable interaction. Considering this, we find it reasonable that the shorter fragment would be more rapidly liberated into the supernatant.

We believe that the labeling of the individual lanes of the gel image is already suitable for readers to comprehend. If the referee could please provide more specific input regarding how the lane labeling should be changed, we would be happy to comply.

4. Is interaction between DDX43 and Ago3 mediated by sDMA modifications?

Several Tudor domain-containing piRNA factors associate with PIWI proteins through sDMA modifications to exert their functions within the piRNA pathway. However, DDX43 has no Tudor domain. Because of this, we assumed that the DDX43–Ago3 interaction may be independent of Ago3-sDMA. We examined this experimentally and found that this was indeed the case: the Ago3 RK mutant lacking the sDMA modification (Nishida *et al.*, 2018) co-immunopurified with DDX43 as the wild-type control did (see the figure below). Thus, the DDX43–Ago3 interaction is independent of Ago3-sDMA.

Referee #2:

In this current work, the authors present data that indicates that the RNA helicase DDX43 is the sought-for protein that releases RNA cleavage products from Ago3. This would be a very worthwhile and important finding. DDX43 is so far not linked to the piRNA pathway, and if true, this finding would answer an obvious question in the piRNA biogenesis field. As such, the findings would be very well suited for EMBO reports.

We thank the referee for these positive comments.

My main three questions/concerns:

1. Where does DDX43 localize in vivo? It is surprising that this rather obvious question is not touched upon in this work. As the authors point out in the text related to Figure 2, piRNA biogenesis is 'enriched' in peri-nuclear structures, generally referred to as nuage. Indeed, many piRNA biogenesis factors (especially those involved in ping-pong) are enriched in nuage (some other factors are found on the mitochondrial surface where 3'

end processing of many piRNAs as well as phased piRNA biogenesis occurs). Vasa for example is a prototypical nuage marker, consistent with its role in releasing the cleavage products from Aubergine/Siwi. It would therefore be very much expected that DDX43 also localizes to nuage. Moreover, nuage localization of DDX43 to nuage should be Ago3 dependent according to the author's model. It would be important to carefully determine and evaluate the subcellular localization of DDX43 in BmN4 cells and to bring these findings into context with the other findings of this paper. If, for example, DDX43 would localize to the nucleus, this would argue against a role of DDX43 in Ago3 biology in vivo.

Unfortunately, the anti-DDX43 monoclonal antibody we raised in this study did not work for immunofluorescence staining. Therefore, we examined the subcellular localization of DDX43 by ectopically expressing Flag-DDX43 in BmN4 cells. This showed that DDX43 was localized in the cytoplasm (see revised Fig 2E). Minor signal was detected at Ago3-positive nuage, as was endogenous Vasa in BmN4 cells (Nishida *et al.*, 2015), but the cytosolic signal was dominant. We thought that this might have been due to the high expression level of Flag-DDX43. Therefore, we tried to reduce the protein level by decreasing the amount of DNA used for transfection. However, this had little effect. We inferred that the data may reflect the actual subcellular localization of endogenous DDX43. Even if this is the case, this does not detract from our claim. For instance, Shutdown, a piRNA biogenesis factor in *Drosophila* ovarian somatic cells (OSCs), was found to have a cytosolic localization when expressed in cultured OSCs instead of localizing to Yb bodies, which are nuage counterparts in OSCs (Hirakata *et al.*, 2019). This clearly indicates that nuage localization can be a criterion in determining the function of a piRNA biogenesis factor but is not mandatory for all factors.

2. A key concern in my opinion is that the in vivo data supporting a role of DDX43 in Ago3 biology is on the weak side. The authors use RNAi to deplete DDX43 in BmN4 cells, and they find pretty much no measurable phenotype when analyzing piRNA populations bound to either Ago3 or Aubergine/Siwi. Given the strong statements in this paper that DDX43 is the factor that is required for Ago3 cleavage product release, this is certainly surprising. The authors mention that this might be caused by incomplete knockdown. That is per se possible, yet not very satisfying. Given that Cas9-mediated knockouts are in principle possible in BmN4 cells (see Tomari lab for example), I strongly suggest that a certified loss of function analysis is required to support the in vivo relevance of DDX43 in the postulated process. I would like to point out that the experiment presented in Figure 3B,C is an interesting way to show that DDX43 might

indeed play a role in release of Ago3 cleavage products (and hence in piRNA biogenesis fueled into Aub). It is, however, a single case experiment and given this result it is surprising that genome wide not even a slight trend in defective Siwi-piRISC formation (ping-pong centered) can be observed (Figure EV2). An alternative approach could be to transition to the Drosophila system where highly potent ways to deplete target proteins via transgenic RNAi exist, and to test here for a (presumably) conserved function of DDX43 in the piRNA ping-pong cycle. I appreciate that this point overall involves quite a bit of effort, yet in my opinion demonstrating a clear in vivo relevance for this protein in the ping-pong cycle would be required.

We have attempted several times to knock out the *DDX43* gene and other genes by applying the CRISPR/Cas9 system to our BmN4 cells but so far we have not had any success. We found that while genomic cleavage occurs successfully every time, the cells do not survive the subsequent cloning step. BmN4 cells appear to dislike being diluted in a culture dish, although such cell dilution is necessary for cloning. We know from our experience that, after CRISPR treatment, cells are quite heterogenic in their genomic sequence, and, therefore, one should avoid using these “mixtures” in subsequent experiments without cloning.

We admit that providing a single case might be a concern. Therefore, we have conducted reporter assays by employing two additional reporter sets and we obtained convincing supportive data. These data are now provided in the revised Fig EV3B.

The fly genetic system is certainly an alternative approach. However, considering that even Vasa function in the ping-pong pathway has not yet been investigated through fly genetics experiments, it would be too much to argue at this moment for the production and analysis of *DDX43* fly mutants. Also, we would like to note that not all functions of piRNA factors are conserved across species. For instance, although the Tudor protein, Papi, is indispensable for piRISC formation in BmN4 cells (Nishida *et al.*, 2018), *Papi* mutant flies are fertile (Liu *et al.*, 2018), which indicates that Papi is unnecessary for the *Drosophila* piRNA pathway.

3. My third point relates to the question whether DDX43 is specific for the cleavage product release from Ago3. The authors mention in their introduction that DDX43 acts on Ago3, yet not on Siwi. I might have overlooked this data, but I cannot find in the entire manuscript a datapoint that shows that addition of DDX43 to Siwi engaged with cleaved targets would not release these. I consider demonstrating that DDX43 is indeed specific to Ago3 as highly relevant. The authors show that such a specificity is indeed the case for Vasa. It only releases cleavage products from Siwi, but not from Ago3.

Such a targeted experiment is required for DDX43 as well in my opinion. If DDX43 is also releasing Siwi's cleavage products, the in vitro ability of this RNA helicase to remove target fragments from an Argonaute might be true in vitro, but not a reality in vivo.

We performed *in vitro* assays and found that DDX43 exhibits little activity toward liberating cleaved RNAs from Siwi. These results are now provided in the revised Fig 3F.

One additional aspect that I consider relevant is to compare DDX43 mutant conditions to Ago3 mutant conditions in terms of RNA-seq as well as piRNA sequencing. How much are Siwi-bound ping-pong piRNAs affected in an Ago3 depleted cell? And if they are strongly affected, would this not speak against a role of DDX43 as an essential release factor for Ago3 given that no changes in ping-pong piRNA levels are seen in DDX43 depleted cells? Moreover, I would consider it important to exclude (or at least to ask for) a potential role of DDX43 in non ping-pong processes. A simple RNAseq experiment on Ago3-depleted versus DDX43 depleted cells should give a good indication whether loss of DDX43 leads to many mis-regulated genes. If that were the case, the authors ideally acknowledge this and build this finding into their discussion.

As we had already noted in the original manuscript (page 11-12), we found no obvious difference between the Siwi-bound piRNA pools before and after DDX43 depletion. Thus, there would appear to be no value in comparing the Siwi-bound piRNA pools between DDX43 depletion and Ago3 depletion. We reasoned that this similarity in pools was because of residual DDX43 remaining in the cells even after extensive RNAi treatment (page 11-12).

Minor comments:

Figure 1E: the vast majority of Ago3 in this experiment is expected to be mature Ago3-RISC, not associated with a target. Why would then DDX43 bind nearly stoichiometrically? Is this interaction also dependent on ATP and is it weaker in Mg²⁺ containing buffers?

We apologize but we do not understand why this stoichiometric interaction is of concern. In the original Fig 3C, the DDX43 mutant, D399A, which failed to interact with ATP, bound Ago3 as well as the wild-type control. This suggested that the Ago3–DDX43 interaction does not require ATP. The Ago3–DDX43 interaction was stronger in the presence of EDTA but weaker in the presence of Mg²⁺ (Fig 1D).

How does the target release kinetics compare between Ago3 and Siwi (DDX43 versus Vasa)?

Because Siwi-bound piRNAs and Ago3-bound piRNAs are totally different in sequences, it is impossible to accurately measure and compare their kinetics. To address this question, we need to have Siwi and Ago3 loaded with an identical, unique piRNA. However, it is currently impossible to prepare such piRISCs.

Dear Mikiko,

Thank you for your patience while your revised manuscript was re-reviewed. We have now received the enclosed reports from the referees. I am happy to say that both referees support publication now, only referee 2 has one minor suggestion that I would like you to incorporate before we can proceed with the official acceptance of your manuscript.

A few other editorial changes will also be required:

- The manuscript has 4 main figures and should therefore be published as a short report. Please combine the results and discussion sections, which is required for short reports.
- Please reduce the number of keywords to 5.
- The reference format needs to be correct to our new Harvard style, listing 10 authors before "et al".
- Please add the Sumitomo Foundation Fiscal 2020 grant number to the manuscript file.
- Please callout the panels of Fig EV2 in the manuscript text.
- Table EV1 needs to be called Dataset EV1 and the legend needs to be added to the first tab. Table EV2 needs to be uploaded as word or excel file.
- The Abstract is missing a heading, the EV figure legends are missing a heading, and the EV table legends should be removed from the Article file.
- The deposited sequencing data needs to be freely available upon online publication of your manuscript.

I attach to this email a related manuscript file with comments by our data editors. Please address all comments in the final manuscript. Especially the figure legends need to provide more detailed information.

With my very best wishes also for the new year!

Esther

Referee #1:

The authors have done an excellent job of revising the manuscript. They have addressed all my concerns. I support its publication.

Referee #2:

Dear Authors.

my apologies for the delayed re-review.

I would like to congratulate the authors for the thorough revision. Most of the raised concerns have been addressed experimentally and the new results further support the author's model.

It is a pity that CRISPR alleles or more potent knockdown conditions cannot be established. In light of this, I suggest adding a sentence to the discussion that specifically discusses this aspect.

Responses

Referee #2:

It is a pity that CRISPR alleles or more potent knockdown conditions cannot be established. In light of this, I suggest adding a sentence to the discussion that specifically discusses this aspect.

We added a sentence, reading “CRISPR system-mediated DDX43 knockout would be desirable to investigate the pronounced phenotype, but was difficult to achieve in BmN4 cells for technical reasons” (page 12).

Editorial Administrator:

- The manuscript has 4 main figures and should therefore be published as a short report. Please combine the results and discussion sections, which is required for short reports.

We re-formatted the whole manuscript as a Report. The Results and the Discussion sections were combined.

- Please reduce the number of keywords to 5.

Done.

- The reference format needs to be correct to our new Harvard style, listing 10 authors before "et al".

Done.

- Please add the Sumitomo Foundation Fiscal 2020 grant number to the manuscript file.

We added the grant number (page 34).

- Please callout the panels of Fig EV2 in the manuscript text.

We callout the panels of Fig EV2 in the manuscript (page 12).

- Table EV1 needs to be called Dataset EV1 and the legend needs to be added to the first tab. Table EV2 needs to be uploaded as word or excel file.

Table EV1 was now called Dataset EV1 (page 8). The legend was added to the first tab. New Table EV1 (used to be Table EV2) is uploaded as a excel file.

- The Abstract is missing a heading, the EV figure legends are missing a heading, and the EV table legends should be removed from the Article file.

We added headings to the Abstract and Fig EV4. The EV table legends were removed from the manuscript.

- The deposited sequencing data needs to be freely available upon online publication of your manuscript.

We will make the deposited sequencing data freely available upon online publication of the paper.

I attach to this email a related manuscript file with comments by our data editors. Please address all comments in the final manuscript. Especially the figure legends need to provide more detailed information.

We addressed all the comments in the final (second revised) manuscript, where modified parts were highlighted in yellow (except re-formatting in the References). The figure legends were modified to have more detailed information as requested.

Prof. Mikiko Siomi
The University of Tokyo
Biological Sciences
2-1-16 Yayoi
Bunkyo-ku, Tokyo 113-0032
Japan

Dear Mikiko,

I am very pleased to accept your manuscript for publication in the next available issue of EMBO reports. Thank you for your contribution to our journal.

I only made very minor changes to the abstract, as the novel findings need to be described in present tense. I hope this is OK, you will see it again in the proofs.

At the end of this email I include important information about how to proceed. Please ensure that you take the time to read the information and complete and return the necessary forms to allow us to publish your manuscript as quickly as possible.

As part of the EMBO publication's Transparent Editorial Process, EMBO reports publishes online a Review Process File to accompany accepted manuscripts. As you are aware, this File will be published in conjunction with your paper and will include the referee reports, your point-by-point response and all pertinent correspondence relating to the manuscript.

If you do NOT want this File to be published, please inform the editorial office within 2 days, if you have not done so already, otherwise the File will be published by default [contact: emboreports@embo.org]. If you do opt out, the Review Process File link will point to the following statement: "No Review Process File is available with this article, as the authors have chosen not to make the review process public in this case."

Should you be planning a Press Release on your article, please get in contact with emboreports@wiley.com as early as possible, in order to coordinate publication and release dates.

Thank you again for your contribution to EMBO reports and congratulations on a successful publication. Please consider us again in the future for your most exciting work.

Best wishes,
Esther

THINGS TO DO NOW:

You will receive proofs by e-mail approximately 2-3 weeks after all relevant files have been sent to our Production Office; you should return your corrections within 2 days of receiving the proofs.

Please inform us if there is likely to be any difficulty in reaching you at the above address at that time. Failure to meet our deadlines may result in a delay of publication, or publication without your corrections.

All further communications concerning your paper should quote reference number EMBOR-2020-51313V3 and be addressed to emboreports@wiley.com.

Should you be planning a Press Release on your article, please get in contact with emboreports@wiley.com as early as possible, in order to coordinate publication and release dates.

Corresponding Author Name: Mikiko C. Siomi

Journal Submitted to: EMBO reports

Manuscript Number: EMBOR-2020-51313V2